# Targeting DNA Double-Strand Break Repair Enhances Radiosensitivity of HPV-Positive and HPV-Negative Head and Neck Squamous Cell Carcinoma to Photons and Protons

**DOI:** 10.3390/cancers12061490

**Published:** 2020-06-07

**Authors:** Eirini Terpsi Vitti, Andrzej Kacperek, Jason L. Parsons

**Affiliations:** 1Cancer Research Centre, Department of Molecular and Clinical Cancer Medicine, University of Liverpool, 200 London Road, Liverpool L3 9TA, UK; E.Vitti@liverpool.ac.uk; 2Clatterbridge Cancer Centre NHS Foundation Trust, Clatterbridge Road, Bebington CH63 4JY, UK; andrzej.kacperek@nhs.net

**Keywords:** ATM, ATR, DNA-PKcs, DNA repair, ionising radiation, proton beam therapy

## Abstract

The response of head and neck squamous cell carcinoma (HNSCC) to radiotherapy depends on human papillomavirus type 16 (HPV) status, and where improved outcome and survival is observed in HPV-positive disease. However, strategies to further radiosensitise the tumours, particularly relatively radioresistant HPV-negative HNSCC, are actively being sought. The impact of targeting the major protein kinases involved in the signaling of DNA double-strand break (DSB) repair, namely ataxia telangiectasia-mutated (ATM), ataxia telangiectasia and Rad3-related (ATR), and the catalytic subunit of DNA-dependent protein kinase (DNA-Pkcs), on the radiosensitisation of HNSCC cells was examined. The response to both conventional photon radiotherapy, but also proton beam therapy, was analysed by clonogenic assays and 3D spheroid growth. We observed that inhibition of ATM, ATR, and particularly DNA-Pkcs, caused a significant reduction in HNSCC cell survival post-irradiation with both photons and protons, with less of an impact on the most radiosensitive HPV-positive cell line. The inhibition of DNA-Pkcs and, to a lesser extent ATM, in combination with radiation was also more effective at inhibiting the growth of 3D spheroids derived from relatively radioresistant HPV-negative HNSCC. Similar effects of the inhibitors were observed comparing photon and proton irradiation, demonstrating the potential for targeting DSB repair as an effective combination treatment for HNSCC.

## 1. Introduction

The incidence of head and neck squamous cell carcinoma (HNSCC) has been reported to be ~800,000 cases per year, and linked with this is the increased rise in oropharyngeal tumours associated with human papillomavirus type 16 (HPV) infection [1,2,3]. It has been clearly demonstrated that patients with HPV-positive squamous cell carcinoma of the oropharynx display improved outcomes and survival rates in comparison to patients with HPV-negative disease [4,5,6,7], which is largely due to the increased responsiveness of HPV-positive tumours to radiotherapy and chemotherapy. Indeed, this difference in radiotherapy response between HPV-positive and HPV-negative HNSCC has been observed in cultured cells derived from patients [8,9,10]. Several studies have indicated that this is caused by defects in the signaling and repair of DNA double-strand breaks (DSBs) in HPV-positive HNSCC cells, largely through the measurement of the DNA damage by neutral comet assays, but also through analysis of surrogate markers, including γH2AX, 53BP1 and RAD51 foci [9,11,12]. However, there are some discrepancies in relation to the specific DSB repair defect, as the reduced expression of proteins involved in both non-homologous end joining (NHEJ; 53BP1 and DNA-Pkcs) and homologous recombination (HR; BRCA2 and RAD51) have been observed. We also recently reported that HPV-positive HNSCC cells have upregulated levels of enzymes involved in the base excision repair (BER) pathway, including XRCC1 and PARP-1 [12]. Furthermore, studies conducted at the genomic level have identified significant genome instability in HPV-positive HNSCC cells and tissues, including alterations in DNA repair genes [13,14,15].

Given that HPV-positive HNSCC cells display an altered capacity for DNA repair, this has revealed that targeting the DNA damage response, particularly in relatively radioresistant HPV-negative HNSCC that display proficient DNA repair mechanisms, may be an effective strategy for the radiosensitisation of the tumour [16]. Specifically, the major protein kinases that co-ordinate the repair of DNA DSBs through NHEJ and HR, namely ataxia telangiectasia-mutated (ATM), ataxia telangiectasia and Rad3-related (ATR), and the catalytic subunit of DNA-dependent protein kinase (DNA-Pkcs), are increasingly being investigated as targets for inhibitors to increase cellular radiosensitisation, principally in response to conventional (photon) radiotherapy. For example, the DNA-Pkcs inhibitors KU0060648 [17] and IC87361 [18], and the ATM inhibitor GSK635416A [19] have been demonstrated to increase radiosensitivity of HNSCC cell lines. A number of studies have also focused on ATR as a target to radiosensitise HNSCC cells, through the inhibitors VE821 [20] and AZD6738 [17,21]. Whilst the majority of these studies have focused on utilising clonogenic assays as an end-point, the ATR inhibitor AZD6738 was shown to impede the growth of 3D spheroids of hypopharyngeal (FaDu) cells in combination with radiation, which are more representative of the original tumour in vivo [21]. Cumulatively, these data would suggest that targeting the DSB repair pathway can be an effective approach for increasing the (photon) radiosensitivity of HNSCC cells.

In addition to conventional (photon) radiotherapy, proton beam therapy is increasingly being utilised for HNSCC treatment [22]. This is due to precise delivery of the radiation dose to the tumour via this radiotherapy technique, resulting in sparing of the normal tissues and organs at risk. However, there is still significant uncertainty regarding the biological impact of protons versus photons, which is important in defining potential combinatorial strategies using targeted drugs to optimise tumour cell radiosensitivity (reviewed in [23]). Specifically, and given that DNA DSBs are the major lesion contributing to ionising radiation-induced cell killing, there are contrasting studies suggesting a dependence on either NHEJ or HR for DNA DSB repair in response to protons. For example, it has been suggested that HR is the major pathway for the repair of DNA DSBs induced in response to protons in A549 lung cancer and glioblastoma cell lines, which would indicate that targeting ATR may be a successful radiosensitisation strategy [24]. However, studies analysing the comparative response of HPV-positive and HPV-negative HNSCC cells to photons versus protons, and the impact of targeting the major kinases involved in DSB repair has not been reported previously. Additionally, utilising HNSCC cells grown as monolayers, but also as 3D spheroids that more accurately reflect the structure and environment of the original tumour, is necessary.

Herein, we have characterised the impact of ATM, ATR and DNA-Pkcs inhibition on the response of HPV-positive and HPV-negative HNSCC cells from the oropharynx to both photons and protons, through the utilisation of clonogenic survival assays and 3D spheroid growth assays. Given that the HPV-negative HNSCC cells are relatively radioresistant compared to their HPV-positive counterparts, we also expanded the results using cells derived from the hypopharynx and oral cavity focusing on 3D spheroid growth, which is more representative of the original tumour and its treatment in vivo. We report that the clonogenic survival and growth of 3D spheroids of cells derived from HPV-positive and HPV-negative HNSCC can be significantly reduced using inhibitors targeting ATM, ATR, and particularly DNA-Pkcs, in combination with both photon and proton irradiation. This suggests that these potential therapeutic strategies could be exploited for the effective treatment of HNSCC, and particularly for relatively radioresistant HPV-negative tumours.

## 2. Results

### 2.1. HPV-Positive HNSCC Cells Are More Radiosensitive than HPV-Negative HNSCC Cells to Photons and Protons

We, and others, have previously demonstrated that there is increased radiosensitivity of cells derived from HPV-positive HNSCC in comparison to HPV-negative HNSCC, which reproduces the effects observed following irradiation of the respective tumours [9,10,12]. To expand on these observations, we used two cell lines derived from each tumour type, where the expression of E6 and E7 oncogenes was confirmed by p16 expression (Figure 1A and Appendix A). Similar to previous data, we were indeed able to reproduce the difference in radiosensitivity of two HPV-positive HNSCC cell lines (UMSCC47 and UPCI-SCC090) in comparison to two HPV-negative HNSCC cell lines (UMSCC6 and UMSCC74A; Figure 1B,C) in response to photon irradiation by clonogenic assays. It should be noted that the colony size was variable between the cell lines, but that colony counting settings were optimised for each cell line and the same settings used across the various treatments for consistency. We also analysed the survival of the same cells following proton irradiation and demonstrated that, similar to results observed following photons, the two most radiosensitive were from HPV-positive HNSCC (Figure 1D,E). The radiosensitivity of the cell lines was generally in the order UMSCC6 > UMSCC74A > UMSCC47 > UPCI-SCC090, and statistical analysis reveals the significantly increased radiosensitivity of UPCI-SCC090 in comparison to UMSCC6 and UMSCC74A (see also Appendix A for linear scale graphs and data fitting).

### 2.2. Survival of HNSCC Cells Following by Photon and Proton Irradiation Can Be Reduced by Targeting ATM, ATR and DNA-Pkcs

Using clonogenic assays, we first analysed the impact of targeting the major protein kinases involved in DNA DSB repair using specific and characterised inhibitors (ATMi, KU-55933; ATRi, VE-821; DNA-Pkcsi, KU-57788) on the survival of HPV-positive and HPV-negative HNSCC incubated with the inhibitors for 24 h in the absence of radiation, versus a vehicle-only control (DMSO). This demonstrated a varied response dependent on the cell line utilised (Appendix A), although ATRi significantly decreased cell survival by 41–54% in all HNSCC cell lines, ATMi by 22–44% in three cell lines (UMSCC6, UMSCC74A and UMSCC47), and DNA-Pkcsi had a significant impact on survival of only one of the four cell lines (UMSCC47) by ~56%. We then analysed the impact of the inhibitors on HNSCC cell survival post-irradiation. As a starting point, we demonstrated that the respective inhibitors, following a 1 h pre-incubation of the cells prior to irradiation, were functional in suppressing ATM, ATR and DNA-Pk phosphorylation, and therefore DSB signaling, in response to photons (Appendix A) and protons (Appendix A). In combination with photon irradiation, we demonstrate that there was a significant impact in reducing cell survival of HPV-negative HNSCC cells in the presence of either ATMi, ATRi or DNA-Pkcsi (1 h pre-incubation, followed by a further treatment for 24 h post-irradiation) versus the DMSO control (Figure 2A–D; see also Appendix A for linear scale graphs and data fitting), with dose enhancement ratios (DER) of 1.91–2.39 (Table 1). The significantly enhanced radiosensitivity of only one HPV-positive HNSCC cell line (UMSCC47) was also seen (Figure 2E–H), although the DER values of 1.36–1.69 were notably lower than those observed in the HPV-negative cells (Table 1). The cell survival of the most inherently radiosensitive HPV-positive cell line (UPCI-SCC090) only appeared to be dramatically decreased in the presence of DNA-Pkcsi (DER of 1.36). These data are supported by statistical analysis (Appendix A) and, in general, DNA-Pkcsi appeared the most potent radiosensitiser of all the HNSCC cell lines.

Following proton irradiation, and similar to photons, we again observed that ATMi and DNA-Pkcsi significantly enhanced the radiosensitisation of both HPV-negative HNSCC cell lines (Figure 3A–D and Appendix A; see also Appendix A for linear scale graphs and data fitting) with DER values of 1.52–2.01 (Table 2). HPV-positive HNSCC cell lines were also radiosensitised, with DER values of 1.24–1.49 (Table 2), following proton irradiation in combination with inhibition of ATM and DNA-Pkcs (Figure 3E–H). However, radiosensitisation was only significantly enhanced in UMSCC47, and not UPCI-SCC090 cell lines (Appendix A). ATRi appeared in general less effective at radiosensitising the HNSCC cells in response to protons (DER values of 1.25–1.48; Table 2).

### 2.3. 3D Spheroid Growth of HNSCC Cells Following by Photon and Proton Irradiation Can Be Inhibited by Targeting ATM, ATR and DNA-Pkcs

We subsequently analysed the impact of DNA DSB repair inhibitors on the radiosensitivity of HNSCC cells utilising 3D spheroids, which more accurately reflect the structure and environment of the original tumour. Of the cells used, unfortunately one HPV-positive cell line (UMSCC47) did not form 3D spheroids that grew during the 15-day analysis period. It was also noted that spheroids from both HPV-negative HNSCC grew significantly faster (peaking at days 8–10 post-seeding) than the one remaining HPV-positive HNSCC (the increase in growth largely occurred at days 7–15 post-seeding). All spheroids grew ~5–8-fold in volume in the absence of any treatments over the analysis period (Figure 4A–I and Appendix A). We demonstrate that ATMi alone caused a significant ~1.7-fold delay in the growth of only HPV-negative HNSCC (UMSCC74A) spheroids, and that the combination of ATMi plus photon irradiation was effective in suppressing the growth of these spheroids by ~2.0-fold compared to radiation alone, but not of the other two spheroid models (Figure 4A–C and Table 3). In contrast, ATRi alone caused a statistically significant ~1.5–1.6-fold growth delay in all spheroid models. The inhibitor significantly exacerbated the effects of photon irradiation, by ~1.3-fold (UPCI-SCC090) to 2.3-fold (UMSCC74A) (Figure 4D–F and Table 3). DNA-Pkcsi alone was, interestingly, ineffective in inhibiting spheroid growth, although the combination of DNA-Pkcsi with photons was effective in suppressing the growth of HPV-negative HNSCC spheroids ~1.4-fold (UMSCC6) and ~1.6-fold (UMSCC74A) compared to the radiation alone (Figure 4G–I).

We observed very similar results in HPV-negative HNSCC spheroids following proton irradiation (Table 3). Here, the combination of protons with ATMi (Figure 5A–C) was significantly effective in only one spheroid model (UMSCC74A) as observed by the ~2-fold growth inhibition versus the radiation alone, whereas ATRi (Figure 5D–F) and DNA-Pkcsi (Figure 5G–I) had a significant impact on delaying the growth of both spheroid models by ~2.5-fold and ~1.9-fold, respectively (see also Appendix A). HPV-positive HNSCC (UPCI-SCC090) spheroids were only significantly radiosensitised, by ~1.6-fold, with protons in the presence of ATRi (Figure 5F). Notably, following both photon and proton irradiation of the HPV-positive UPCI-SCC090 spheroids, there was a reduced impact of the inhibitors compared to the radiation alone, which is consistent with this being the most radiosensitive cell line, as observed by clonogenic assays (Figure 1B,D).

We extended our observations of the effectiveness of inhibitors targeting ATM, ATR and DNA-Pkcs in radiosensitising oropharyngeal HNSCC cells by utilising additional spheroid models from HPV-negative HNSCC, which are relatively more radioresistant than HPV-positive HNSCC. These were designed to gain further evidence that DNA DSB repair inhibition can enhance the impact of photons and protons in preventing spheroid growth, which are more representative of the original tumour and its treatment in vivo. We therefore used spheroids from FaDu and A253 cell lines that originate from the hypopharynx and oral cavity, respectively, of which we observed that these increased dramatically in volume (by ~50-fold and ~15-fold, respectively) over a period of 15 days post-seeding (Figure 6A–L and Appendix A). FaDu spheroids were particularly resistant to ATMi, ATRi and DNA-Pkcsi alone, as observed by the lack of impact on spheroid growth. The A253 spheroids appeared to display some delayed growth in the presence of the inhibitors alone, particularly at the 12- and 15-day time points, although this was not statistically significant across the whole time course (Table 4). The combination of photons with either of the inhibitors significantly suppressed the growth of A253 spheroids, which was markedly enhanced by ~2.8–3.2-fold versus the radiation alone (Figure 6A–F and Table 4). FaDu spheroids were only significantly radiosensitised in the presence of DNA-Pkcsi following photon irradiation, through a dramatic ~4.6-fold decrease in spheroid growth. In response to proton irradiation, ATRi was not significantly effective at radiosensitising the cells, but the combination of ATMi with protons was able to suppress growth of both A253 and FaDu spheroids by ~3.7-fold. Furthermore, DNA-Pkcsi was particularly effective in combination with protons as observed by the ~3.6-fold and ~7.6-fold decrease in the spheroid growth of A253 and FaDu cells, respectively, in comparison to radiation alone (Figure 6G–L and Table 4).

## 3. Discussion

Accumulating evidence has suggested that the increased response of patients with HPV-positive versus HPV-negative HNSCC to radiotherapy, and thus the improved survival rates, is caused by defects in the repair of DNA DSBs [9,11,12]. Therefore, targeting key enzymes involved in DNA DSB repair, particularly the protein kinases ATM, ATR and DNA-Pkcs, in relatively radioresistant HPV-negative HNSCC that are DSB repair-proficient is considered to be an approach to sensitise these tumours to radiotherapy. Indeed, there is evidence of at least one clinical trial utilising either ATRi or DNA-Pkcsi in combination with conventional radiotherapy that is currently underway [16]. Additionally, while there is an increasing use of proton beam therapy for the treatment of HNSCC, there is no preclinical evidence to date examining the impact of DNA DSB repair inhibitors in combination with protons, and whether there is any substantial difference compared to that observed following photon irradiation. In this study, we have now analysed the effect of ATMi, ATRi and DNA-Pkcsi on both monolayer and 3D spheroid models of HPV-positive and HPV-negative HNSCC in combination with photons and protons.

Interestingly, we discovered that targeting either ATM, ATR or DNA-Pkcs can decrease the clonogenic survival of HNSCC cells in response to photons and protons. DNA-Pkcsi appeared particularly effective in all cell lines in combination with radiation. This would correlate with studies in HPV-negative HNSCC cells describing downregulation of DNA-Pkcs using siRNA in UTSCC15 and UTSCC45 cells [25], as well as the DNA-Pkcs inhibitors KU0060648 in HN4 and HN5 cells [17], and IC87361 in UTSCC54, UTSCC74B and UTSCC76B cells [18], which were shown to enhance radiosensitisation. Only a single study has examined ATM inhibition (GSK635416A) in HNSCC cells [19], although this demonstrated increased radiosensitivity in five HPV-negative HNSCC cell lines (UTSCC2, UTSCC8, UTSCC24A, UTSCC36 and UTSCC40), which is comparable with our data. However, there are a number of studies that have focused on ATR as a target, including using siRNA in UPCI-SCC029B, UPCI-SCC040 and UPCI-SCC131 cells [26]. Additionally, the ATR inhibitor VE821 displayed improved radiosensitivity in SQ20B cells [20], and an alternative inhibitor, AZD6738, showed the same phenotype in Cal27, FaDu, HN4 and HN5 cells [17,21]. In our experiments utilising clonogenic assays, ATRi appeared to be less effective at radiosensitising cells following proton irradiation. We also observed less of an impact of DNA DSB repair inhibition in combination with radiation in HPV-positive HNSCC cells, particularly the UPCI-SCC090 cell line largely as this is the most inherently radiosensitive as shown here, and in our previous study [12].

Utilising 3D spheroid models that more accurately replicate the structure and environment of the original tumour, we further demonstrated the effectiveness of DNA-Pkcsi in combination with both photons and protons in inhibiting growth of all the HPV-negative HNSCC spheroids analysed. Interestingly though, inhibition of DNA-Pkcs alone did not appear to have any impact on the growth of 3D spheroids of both HPV-positive and HPV-negative HNSCC (which was largely supported by utilising clonogenic survival assays). This suggests that DNA-Pkcs is not essential for HNSCC cell growth and survival in the absence of ionising radiation-induced stress. Nevertheless, and similar to clonogenic assay results, the combination strategy of DSB inhibition (particularly ATMi and DNA-Pkcsi) did not significantly enhance the effect of radiation on the HPV-positive HNSCC spheroids (UPCI-SCC090), due to these cells being the most radiosensitive. The inhibition of ATR displayed some effectiveness in combination with photons and protons in preventing spheroid growth. However, less of an impact was observed on the relatively radioresistant HPV-negative HNSCC spheroid models, FaDu and A253, that displayed significant spheroid growth over the time period post-irradiation. This observation is similar to previous data utilising the ATR inhibitor AZD6738 with photons only, which demonstrated that this combination did not impede growth of 3D spheroids of FaDu cells [21]. Noteworthily, as a monotherapy, the inhibition of ATR alone in the absence of radiation was effective in inhibiting clonogenic survival, but also the growth of HNSCC spheroids (apart from FaDu and A253), which was comparable to the impact caused by a single dose of radiation alone.

Cumulatively, our results suggest that targeting DNA DSB repair via NHEJ (ATM and DNA-Pkcs) or HR (ATR) can exacerbate the impact of photons in radiosensitising HNSCC cell models, and that the combination of DNA-Pkcsi with photons in HPV-negative HNSCC cells that are relatively radioresistant was particularly effective. This adds to the growing preclinical evidence [17,18,19,20,21,25,26] that this is an effective combination for the treatment of HNSCC that should be investigated further, particularly using more advanced 3D models (e.g., patient-derived organoids) and appropriate in vivo experiments. However, we now also demonstrate that DSB repair inhibition, particularly DNA-Pkcsi and to a lesser extent ATMi, are efficient in reducing the survival and spheroid growth of HNSCC cells in response to protons. In fact in general, relatively similar results were observed comparing photons and protons, although the DER values derived from clonogenic assay results were much lower with ATRi following protons than with photons. This would contradict some very limited evidence suggesting a greater dependence on the HR pathway mediated by ATR for repairing DNA DSBs induced by protons, which was obtained using RAD51 siRNA in A549 lung cancer cells [24]. In fact other studies, largely conducted in Chinese hamster ovary cells, reflect that NHEJ, coordinated by ATM and DNA-Pkcs, is the major DSB repair pathway employed following proton irradiation [27,28]. This is in agreement with our results. Consequently, we would advocate that inhibition of NHEJ through DNA-Pkcs is the most promising strategy in optimising the radiosensitisation of HNSCC cells with either photons or protons. Nevertheless, it should be noted that our study utilised low linear energy transfer (LET) protons at the entrance dose of a pristine beam, and that different results may be obtained with cells irradiated at or around the Bragg peak where the LET increases. This is due to the increased amount of complex DNA damage, where multiple lesions are generated in close proximity, and therefore the potential for the generation of complex DNA DSBs that could have a different requirement for either NHEJ or HR [23]. We are also acutely aware of the availability of more potent and selective inhibitors than the ones used in the current study, specifically those targeting ATM (e.g., AZD1390), ATR (e.g., AZD6738) and DNA-Pkcs (e.g., AZD7648), which require examination of their potential to radiosensitise HNSCC cell models following photon and proton irradiation. These points are consequently the subject of our ongoing and future studies.

## 4. Materials and Methods

### 4.1. Cell Lines and Culture Conditions 

Oropharyngeal squamous cell carcinoma cells (UMSCC6, UMSCC74 and UMSCC47) were kindly provided by Prof T. Carey, University of Michigan, USA. Cells from the hypopharynx (FaDu) and submaxillary gland (A253) originated from ATCC (Teddington, UK). HPV-positive oropharyngeal squamous cell carcinoma cells (UPCI-SCC090) were kindly provided by Dr S. Gollin from the University of Pittsburgh. All cells, apart from UPCI-SCC090 and FaDu (which were cultured in Minimal Essential Medium (MEM)), were routinely cultured as monolayers in Dulbecco’s Modified Eagle Medium (DMEM) supplemented with 10% fetal bovine serum, 2 mM L-glutamine, 1× penicillin-streptomycin and 1× non-essential amino acids. All cells were cultured under standard conditions in 5% CO2 at 37 °C, and were authenticated in our laboratory by short tandem repeat (STR) profiling.

### 4.2. Clonogenic Assays 

Cells were harvested and a defined number seeded in triplicate into 6-well plates or 35 mm dishes before incubation overnight in 5% CO_2_ at 37 °C to allow the cells to attach. Plating efficiencies for the cells were as followed: UMSCC6 (~10%), UMSCC74A (~10%), UMSCC47 (~10%) and UPCI-SCC090 (~2%). For inhibition experiments, cells were pretreated with DMSO (as a vehicle only control), 10 µM ATM inhibitor (ATMi; KU-55933), 1 µM ATR inhibitor (ATRi; VE-821) or 1 µM DNA-Pkcs inhibitor (DNA-Pkcsi; KU-57788; Selleck Chemicals, Munich, Germany) for 1 h prior to irradiation. Cells were then irradiated using a CellRad x-ray irradiator (Faxitron Bioptics, Tucson, AZ, USA) or with a passive scattered horizontal proton beam line of 60 MeV maximal energy, as previously described [29,30]. Higher doses of protons were comparatively used due to cells being positioned at the entrance dose of a pristine (unmodulated) beam (~1 keV/µm). Following irradiation, fresh media containing inhibitors was added to the cells for 24 h, which was then replaced with fresh media alone and colonies allowed to grow for 7–12 days, prior to fixing and staining with 6% glutaraldehyde and 0.5% crystal violet for 30 min. Dishes were washed, left to air dry overnight and colonies counted using the GelCount colony analyser (Oxford Optronics, Oxford, UK). Colony counting settings were optimised for each cell line, based on inclusion of distinct colonies of specific size and intensity, although the same settings were used across the various treatments. Relative colony formation (surviving fraction) was expressed as colonies per treatment level versus colonies that appeared in the untreated control, and data was derived from at least three individual biological replicates.

### 4.3. Spheroid Growth Assays

Cells (500–1000/well) were seeded in triplicate in 100 µL Advanced MEM media (Life Technologies, Paisley, UK) containing 1% B27 supplement, 0.5% N-2 supplement, 2 mM L-glutamine, 1× penicillin-streptomycin, 5 µg/mL heparin, 20 ng/µL epidermal growth factor and 10 ng/µL fibroblast growth factor into 96-well ultra-low attachment plates (Corning B.V. Life Sciences, Amsterdam, The Netherlands) and spheroids of ~200 µm in diameter allowed to form for 48 h (Day 3). DMSO, ATMi, ATRi and DNA-Pkcsi were added 1 h prior to irradiation. Post-irradiation, 50 µL media was removed and replaced with 50 μL fresh media containing DMSO or inhibitors for 24 h, and then 50 µL media removed and replaced by 100 μL with fresh media alone. Images of spheroids were captured up to 15 days post-seeding using an EVOS M5000 Imaging System (Life Technologies, Paisley, UK). The diameter of the spheroids was analysed using ImageJ, and used to calculate spheroid volume using the formula 4/3 × π × (d/2)^3^.

### 4.4. Statistical Analysis

Dose enhancement ratios (DER) were used to assess the significance of the clonogenic assay results. DER values are derived from the ratio of the dose (Gy) required for a surviving fraction of 0.5 in the vehicle (DMSO) treated cells (D50_DMSO_), over the dose (Gy) required for the same surviving fraction in the inhibitor treated cells (D50_inhibitor_) [DER = D50_DMSO_/D50_inhibitor_]. D50 values were calculated using a linear quadratic fitting on each curve. Statistical analysis of spheroid growth data was performed on the dataset across the 15-day growth period using a one-way ANOVA. For this, the effect of each inhibitor on the spheroid growth was compared against the vehicle (DMSO) for a given radiation dose and radiation type. *p*-values of <0.05 highlight statistical significance between DMSO and inhibitor treated spheroids over the growth period. 

## 5. Conclusions

We have demonstrated that the inhibition of DNA DSB repair can effectively act in combination with both conventional (photon) radiotherapy and proton beam therapy in radiosensitising in vitro models of HNSCC. DNA-Pkcsi was shown to be particularly effective in preventing clonogenic survival and 3D spheroid growth of HNSCC, and specifically models of relatively radioresistant HPV-negative HNSCC. Our data suggest that targeting DNA-Pkcs in combination with radiotherapy can be an effective strategy for the treatment of HNSCC.

## Figures and Tables

**Figure 1 cancers-12-01490-f001:**
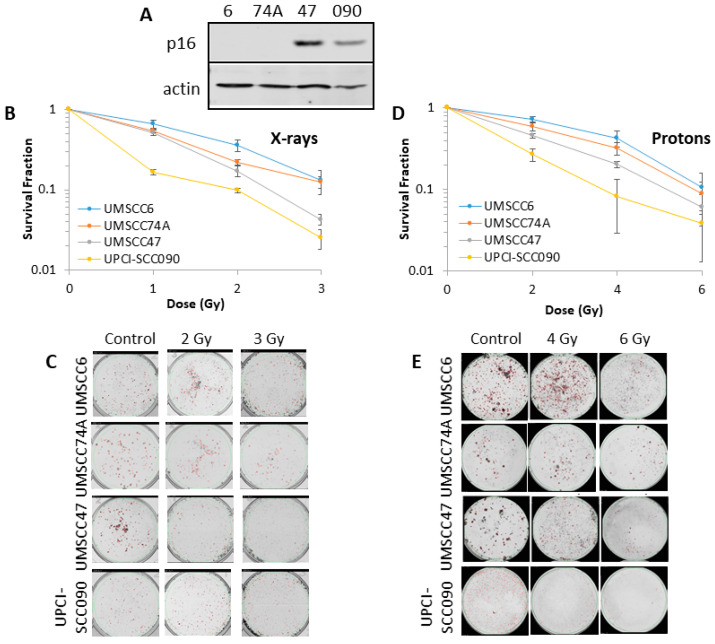
Comparative radiosensitivity of human papillomavirus type 16 (HPV)-negative and HPV-positive head and neck squamous cell carcinoma (HNSCC) cells in response to photons and protons. (**A**) Whole cell extracts from HNSCC cells were prepared and analysed by immunoblotting with the indicated antibodies. Clonogenic survival of HNSCC cells following treatment with increasing doses of (**B**,**C**) x-rays or (**D**,**E**) protons was analysed from three to four biologically independent experiments. (**B**,**D**) Shown is the surviving fraction ± S.E. (**C**,**E**) Representative images of colonies in non-irradiated and irradiated plates (the latter were seeded with four times and eight times the number of cells, accordingly). Statistical analysis using a one sample *t*-test of surviving fractions at a 2 Gy dose of x-rays reveals significant differences of *p* < 0.03 (UMSCC6 vs. UPCI-SCC090), *p* < 0.005 (UMSCC74A vs. UPCI-SCC090); and at a 4 Gy dose of protons of *p* < 0.04 (UMSCC6 vs. UPCI-SCC090), *p* < 0.04 (UMSCC74A vs. UPCI-SCC090). The uncropped blots and molecular weight markers of Figure 1 are shown in Appendix A.

**Figure 2 cancers-12-01490-f002:**
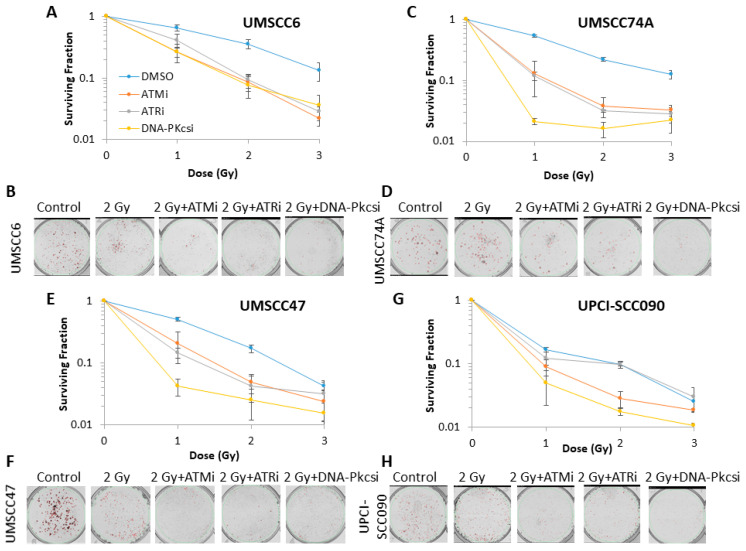
Inhibition of ataxia telangiectasia-mutated (ATM), ataxia telangiectasia and Rad3-related (ATR) and DNA-dependent protein kinase (DNA-Pkcs) can enhance sensitivity of HNSCC cells to photon irradiation. Clonogenic survival of HNSCC cells following treatment with increasing doses of x-rays in the presence of DMSO (Control), ATMi (10 µM), ATRi (1 µM) and DNA-Pkcsi (1 µM) was analysed from three biologically independent experiments. (**A**,**C**,**E**,**G**) Shown is the surviving fraction ± S.E. (**B**,**D**,**F**,**H**) representative images of colonies in non-irradiated and irradiated plates (the latter of which were seeded with four times the number of cells).

**Figure 3 cancers-12-01490-f003:**
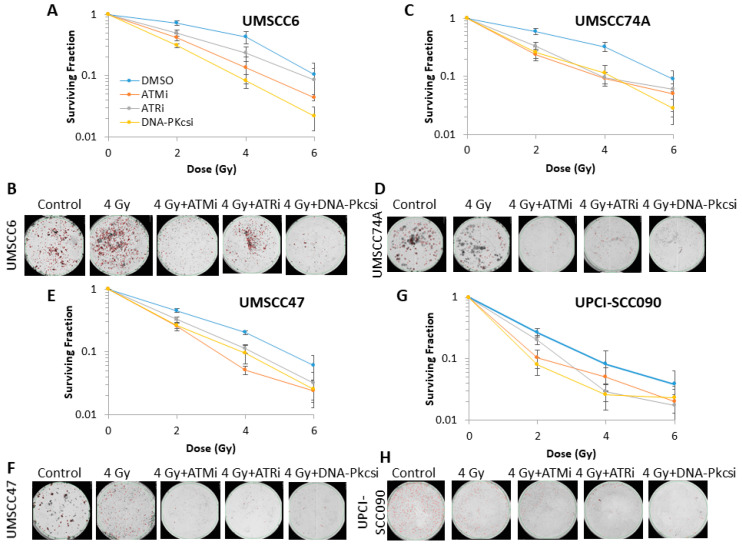
Inhibition of ATM, ATR and DNA-Pkcs can enhance sensitivity of HNSCC cells to proton irradiation. Clonogenic survival of HNSCC cells following treatment with increasing doses of protons in the presence of DMSO (Control), ATMi (10 µM), ATRi (1 µM) and DNA-Pkcsi (1 µM) was analysed from four biologically independent experiments. (**A**,**C**,**E**,**G**) Shown is the surviving fraction ± S.E. (**B**,**D**,**F**,**H**) representative images of colonies in non-irradiated and irradiated plates (the latter of which were seeded with four times the number of cells).

**Figure 4 cancers-12-01490-f004:**
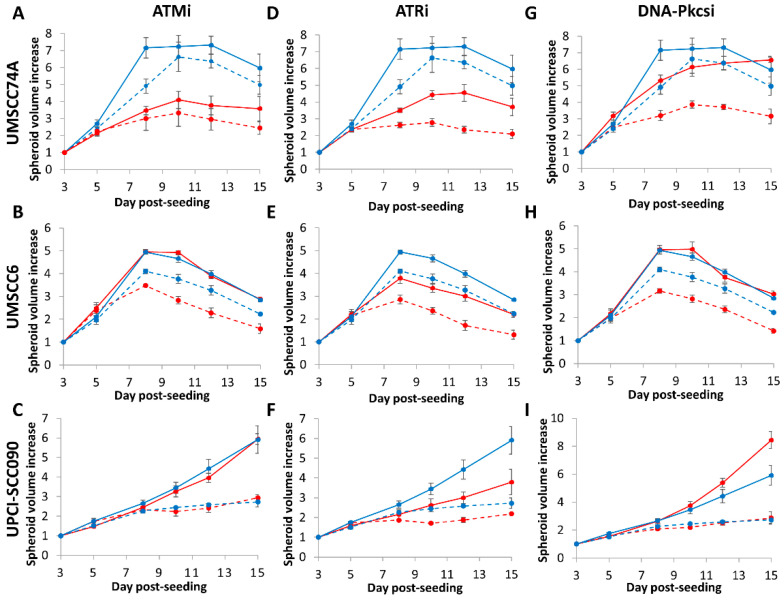
Inhibition of ATM, ATR and DNA-Pkcs in combination with photons can decrease growth of HNSCC 3D spheroids. Spheroids were allowed to develop for 48 h, pretreated with DMSO (Control), ATMi (10 µM), ATRi (1 µM) and DNA-Pkcsi (1 µM), and irradiated with a single dose (1 Gy) of x-rays. Spheroid growth of (**A**,**D**,**G**) UMSCC74A, (**B**,**E**,**H**) UMSCC6 and (**C**,**F**,**I**) UPCI-SCC090 was measured by microscopy and analysed from three biologically independent experiments. Solid blue line is DMSO only, dashed blue lines are DMSO plus 1 Gy x-rays, solid red line is inhibitor only, dashed red lines are inhibitors plus 1 Gy x-rays. Shown is the spheroid volume ± S.E.

**Figure 5 cancers-12-01490-f005:**
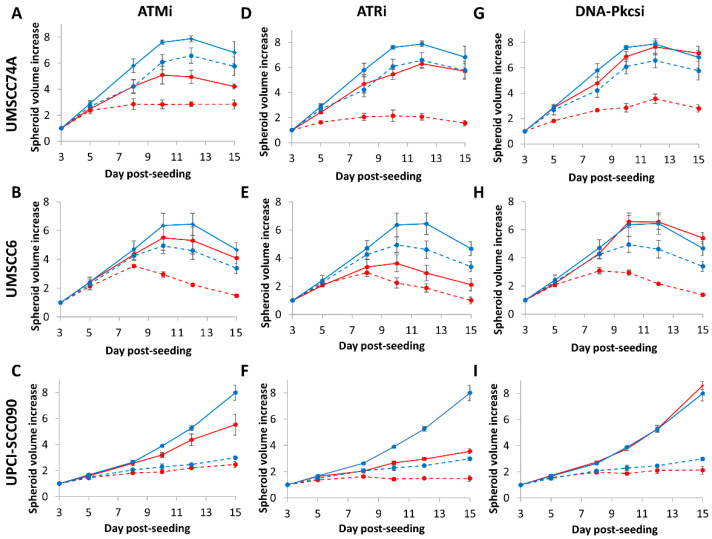
Inhibition of ATM, ATR and DNA-Pkcs in combination with protons can decrease growth of HNSCC 3D spheroids. Spheroids were allowed to develop for 48 h, pretreated with DMSO (Control), ATMi (10 µM), ATRi (1 µM) and DNA-Pkcsi (1 µM), and irradiated with a single dose (2 Gy) of protons. Spheroid growth of (**A**,**D**,**G**) UMSCC74A, (**B**,**E**,**H**) UMSCC6 and (**C**,**F**,**I**) UPCI-SCC090 was measured by microscopy and analysed from three biologically independent experiments. Solid blue line is DMSO only, dashed blue lines are DMSO plus 2 Gy protons, solid red line is inhibitor only, dashed red lines are inhibitors plus 2 Gy protons. Shown is the spheroid volume ± S.E.

**Figure 6 cancers-12-01490-f006:**
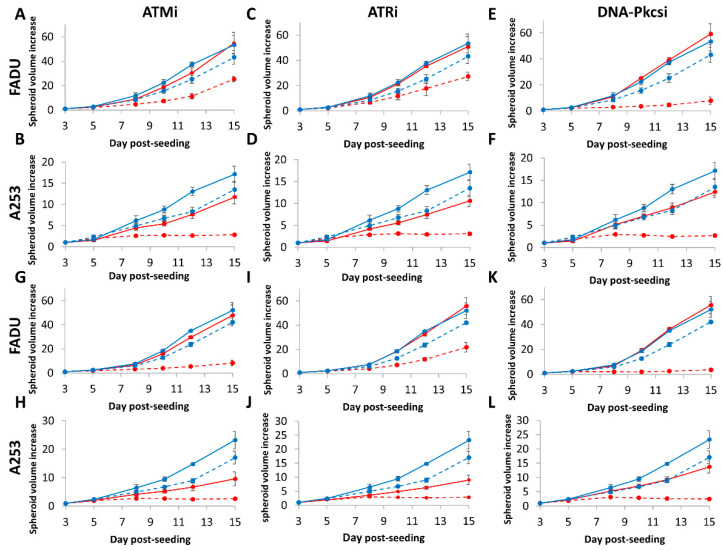
Inhibition of ATM, ATR and DNA-Pkcs in combination with photons and protons can decrease growth of HPV-negative HNSCC 3D spheroids. Spheroids were allowed to develop for 48 h, pretreated with DMSO (Control), ATMi (10 µM), ATRi (1 µM) and DNA-Pkcsi (1 µM), and irradiated with a single dose of (**A**–**F**) x-rays at 1 Gy or (**G**–**L**) protons at 2 Gy. Spheroid growth of (**A**,**C**,**E**,**G**,**I**,**K**) hypopharynx (FaDu) and (B, D, F, H, J and L) A253 was measured by microscopy and analysed from three biologically independent experiments. Solid blue line is DMSO only, dashed blue lines (**A**–**F**) are DMSO plus 1 Gy x-rays or (**G**–**L**) 2 Gy protons, solid red lines are inhibitor only, dashed red lines are inhibitor plus (**A**–**F**) 1 Gy x-rays or (**G**–**L**) 2 Gy protons. Shown is the spheroid volume ± S.E.

**Table 1 cancers-12-01490-t001:** Dose enhancement ratios calculated at 50% cell survival (DER) following ATM, ATR and DNA-Pkcs inhibition versus DMSO controls in HNSCC cells in response to photons.

Inhibitor	UMSCC6	UMSCC74A	UMSCC47	UPCI-SCC090
ATM	2.06	1.91	1.38	1.15
ATR	1.91	2.01	1.36	1.02
DNA-Pkcs	1.93	2.39	1.69	1.36

**Table 2 cancers-12-01490-t002:** Dose enhancement ratios calculated at 50% cell survival (DER) following ATM, ATR and DNA-Pkcs inhibition versus DMSO controls in HNSCC cells in response to protons.

Inhibitor	UMSCC6	UMSCC74A	UMSCC47	UPCI-SCC090
ATM	1.62	1.52	1.49	1.24
ATR	1.25	1.42	1.28	1.30
DNA-Pkcs	2.01	1.64	1.38	1.32

**Table 3 cancers-12-01490-t003:** Targeting of ATM, ATR and DNA-Pkcs alone and in combination with photons and protons to decrease 3D HPV-positive and HPV-negative HNSCC spheroid growth.

Inhibitor	UMSCC74A	UMSCC6	UPCI-SCC090
ATM	*p* < 0.0002	*p* = 0.60	*p* = 0.34
ATR	*p* < 0.003	*p* < 0.002	*p* < 0.006
DNA-Pkcs	*p* = 0.59	*p* = 0.89	*p* = 0.54
ATM + photons	*p* < 0.004	*p* = 0.18	*p* = 0.76
ATR + photons	*p* < 0.0005	*p* < 0.02	*p* < 0.03
DNA-Pkcs + photons	*p* < 0.02	*p* < 0.05	*p* = 0.08
ATM + protons	*p* < 0.02	*p* = 0.06	*p* = 0.24
ATR + protons	*p* < 0.0002	*p* < 0.003	*p* < 0.0008
DNA-Pkcs + protons	*p* < 0.03	*p* < 0.02	*p* = 0.18

Statistical analysis was performed on all the dataset across the 15-day growth period using a one-way ANOVA, comparing the growth of inhibitor treated spheroids against the appropriate DMSO control (± radiation).

**Table 4 cancers-12-01490-t004:** Targeting of ATM, ATR and DNA-Pkcs alone and in combination with photons and protons to decrease 3D HPV-negative HNSCC spheroid growth.

Inhibitor	FaDu	A253
ATM	*p* = 0.69	*p* = 0.49
ATR	*p* = 0.89	*p* = 0.72
DNA-Pkcs	*p* = 0.82	*p* = 0.88
ATM + photons	*p* = 0.09	*p* < 0.002
ATR + photons	*p* = 0.28	*p* < 0.003
DNA-Pkcs + photons	*p* < 0.003	*p* < 0.002
ATM + protons	*p* < 0.03	*p* < 0.006
ATR + protons	*p* = 0.24	*p* = 0.11
DNA-Pkcs + protons	*p* < 0.005	*p* < 0.002

Statistical analysis was performed on the dataset across the 15-day growth period using a one-way ANOVA, comparing the growth of inhibitor treated spheroids against the appropriate DMSO control (± radiation).

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
