# Peer review of "Targeting DNA Double-Strand Break Repair Enhances Radiosensitivity of HPV-Positive and HPV-Negative Head and Neck Squamous Cell Carcinoma to Photons and Protons"

_cancers, 2020, doi:10.3390/cancers12061490_

Round 1

Reviewer 1 Report

Authors' rebuttal is shown, followed by critique of revised manuscript, marked by ///

Reviewer 1
The paper needs to be very substantially modified prior to re-review. The following comments are provided to help the authors make the paper better: - Statistical analysis of the responses to treatments needs to be provided. As it stands, figures are presented on a semi-log scale, which magnifies small quantitative differences and compresses large differences. Thus minor differences between cell lines, in response to various combinations of treatments, is highlighted. The data need to be analyzed for statistical significance.

Response: We thank the Reviewer for their comment regarding statistical analysis. Clonogenic assay data (Figures 1-3) are indeed presented on a logarithmic scale, which is a routine and standard way of displaying this type of data. We had already provided the dose enhancement ratios (DER) on these data (T ables 1-2), demonstrating that the addition of the drugs are enhancing cellular radiosensitisation. However in response to the Reviewer’s comment, we have now added statistical analysis of the clonogenic data at intermediate dose of photons (Table S1) and protons (Table S2), which further demonstrate significantly enhanced radiosensitisation of cells (apart from UPCI- SCC090 with protons) with either ATM, ATR or DNA-Pkcs inhibition. The appropriate changes have be made to the text (e.g. within lines 127-135).

//// Please add the clonogenic assay data (Figs. 1 – 3), graphed on a linear scale, as additional figures; either log scale or linear scale graphs should be presented in the main paper, and the other should be presented in supplement.  Please indicate, on the linear graphs, where the dose enhancement ratios are being calculated. Presumably the authors are interpolating between radiation doses to calculate the DERs, but they refer to inhibitor treatments, whose doses are not varied here. Please explain this apparent discrepancy.  Are the DERs significantly different in comparing the HPV-positive vs HPV-negative lines?  In Tables 1 and 2, why are DERs presented for a single dose of drug inhibitors, whereas the variable being modified (Figures 1 and 2) is the radiation dose ? Shouldn’t the DER be calculated for the drugs?

- Which doses or time points are taken for comparisons between cell lines? This represents another limitation of quantitation in the paper which reduces ability to interpret results.

Response: The DER values included in the original submission (Tables 1-2) on the clonogenic assay data were derived from the ratio of the dose required for a surviving fraction of 0.5 in the DMSO versus the inhibitor-treated cells, and which was indicated in the titles of the Tables. For the 3D spheroid data (Figures 4-5), the statistical analysis was performed using ANOVA on the whole dataset across the 15-day growth period comparing DMSO and inhibitor-treated cells (±radiation). In response to the Reviewer’s comment, we have now added an additional section in the Materials and Methods to clarify this (section 4.4. Statistical analysis), as well as the appropriate footnotes to Tables 3-4.

/// Again, clonogenic assay data reflect various radiation doses, but the DER values are presented for inhibitors (whose doses weren’t varied as far as I see in these figures).  It is not clear to me what the authors intend for the reader to glean from the DER values presented in Tables 1 and 2. Is there a significant difference between HPV-positive vs -negative cell lines? Are the clonogenic assay data themselves (as presented in the graphs) significantly different?

- Tables presenting p-values don't show the quantitative fold change or percent of cells surviving. These quantitative data need to be added.

Response: We appreciate the Reviewer’s comment. The p values in Tables 3-4 do indeed only demonstrate statistical significance of the impact of the inhibitors on 3D spheroid growth. Therefore, we have now added fold changes in growth inhibition through the text (lines 170-185 and 222-229) which provide a more quantitative aspect to our data.

/// I don’t see any fold changes in growth added in the Tables (which is preferred), nor in the text at the indicated lines. This is unacceptable.

- Additional cell lines are included at the end of the paper, but they are not studied in early figures. Why was this organized in this manner? All cell lines should be mentioned and studied throughout the study.

1

Response: We appreciate the Reviewer’s suggestion. Our study was originally focussed on four cell lines, two HPV-negative (UMSCC74A and UMSCC6) and two HPV-positive (UMSCC47 and UPCI- SCC090) derived from the oropharynx. This follows on from our previously published data (Nickson et al., 2017, Oncotarget), where we fully characterised the DNA damage response of the cells following x-ray irradiation. In this study, we have now characterised the impact of ATM, ATR and DNA-Pkcs inhibition on the response of the same cells to both photons and protons, through the utilisation of clonogenic survival assays and 3D spheroid growth assays. Given that the HPV- negative cells are relatively radioresistant compared to their HPV-positive counterparts, we specifically expanded the results using FaDu and A253 cells (from hypopharynx and oral cavity, respectively) focussing on 3D spheroid growth which is more representative of the original tumour, and its treatment, in vivo. It is clear from this data that inhibition of particularly DNA-Pkcs, can also be used as a strategy to radiosensitise these additional models to photons and protons.

/// please direct these comments to the reader of the paper, not to reviewer. – Add the rationale to the Introduction of the paper.

- Acronyms DNA-PKcs and DNA-PKcsi are not defined. Similarly, other gene names followed by I are not defined.

Response: We thank the Reviewer for their suggestion, and have now defined all acronyms in the text (e.g. lines 18, 54-55 and 111-112).

/// OK

- References to genomics paper about HPV-positive head and neck cancer cell lines and tumors should be included in the citations, such as doi: 10.1101/gr.164806.113 ; DOI: 10.1158/1078- 0432.CCR-14-1101; PMID: 30563911. These papers address mutations affecting specific cell lines and disrupting DNA repair pathways.

Response: We appreciate the Reviewer’s suggestion, and have now added the papers into the Introduction (lines 46-48).

/// OK

- Choices of inhibitors don't match what is being studied currently in clinical trials. Can the authors explain why they chose not to investigate the particular drugs currently tested in patients ?

Response: We thank the Reviewer for their comment. We initiated the study utilising the chosen inhibitors given that they were well studied and characterised. However, we are fully aware of the availability of now more potent inhibitors of ATM (AZD1390), ATR (AZD6738) and DNA-Pkcs (AZD7648), which are in clinical trials. Interestingly, only AZD6738 is currently in a presurgical trial in HNSCC. We are currently in the process of investigating these more up to date inhibitors as a follow-up to the current study, which has been acknowledged in the Discussion (lines 317-321).

/// OK

/// OK

- images of stained cells show some concerning artifacts. Some wells show stained cell colonies and/ or a haze near the well center, not seen uniformly at the well periphery. Can the authors explain this?

2

Response: We appreciate the Reviewer’s comment. The staining at the centre of a very small number of wells is not quite uniform due to the increased numbers of colonies growing in this region, and doesn’t relate to any artifacts.

/// I see many wells in Figs. 1, 2 and 3 showing this non-uniform clustering of cells. Please address this in the paper for readers, not for reviewer. Also there is at least one identical well in Figs. 1 and 2 (control for 2 Gy treatment) that makes me question how many technical replicates were performed. Please state this in the figure legends for the reader. Are these “representative” wells shown in figures?

- quantitation of the stained cell colonies doesn't appear to match the values shown in the corresponding graphs. How exactly did the authors quantify the cell colonies -- via total density of stain in the well,, by counting discrete colonies regardless of their size, etc. ? The quantitation of the cells is crucial for the paper, and this is unclear.

Response: We thank the Reviewer for their comment. Colonies were counted using the software provided with the GelCount colony counter from Oxford Optronics. Here, the counting settings were optimized for each cell line (given the difference in colony formation and growth between these) to ensure that only distinct colonies were counted. Importantly, and to exclude any bias, the same settings were then used to count across the inhibitor treated colonies in the absence and presence of radiation (photons and protons) to ensure consistency across these. This has now been included into the Materials and Methods (lines 349-351).

/// OK

Reviewer 2 Report

Major concerns:

  • Figure 1B-D:  No statistical analysis of the data is provided for the results shown in this figure.
  • How do the authors reconcile molecular data shown in FigS2, and in particular the effects of inhibitors in non-irradiated cells (roughly identical levels of phosphorylation of ATM, ATR and Pkcs in control cells and cells treated with inhibitors alone) with their biological effects described in FigS1, FigS3 and Fig5?
  • The authors claim the "though, inhibition of DNA-Pkcs alone did not appear to have any impact on the growth of 3D spheroids of both HPV-positive and HPV-negative HNSCC (which was largely supported by utilizing clonogenic survival assays), suggesting that DNA-Pkcs is not essential for HNSCC cell growth and survival in the absence of ionizing radiation-induced stress". However, the western blots shown in FigS2 do not provide convincing evidence that Pkci is effective in inhibiting Pkc phosphorylation on non-irradiated cells

Reviewer 3 Report

I do not have any specific comment or suggestion.

Reviewer 4 Report

No further comments.

Round 2

Reviewer 1 Report

Please carefully edit the manuscript. I found several simple grammatical errors that should be corrected. 

Author Response

Please carefully edit the manuscript. I found several simple grammatical errors that should be corrected.

Response: We have now been through the manuscript in full detail, and corrected any spelling and grammatical errors where necessary.

This manuscript is a resubmission of an earlier submission. The following is a list of the peer review reports and author responses from that submission.

Round 1

Reviewer 1 Report

The paper needs to be very substantially modified prior to re-review. The following comments are provided to help the authors make the paper better: - Statistical analysis of the responses to treatments needs to be provided. As it stands, figures are presented on a semi-log scale, which magnifies small quantitative differences and compresses large differences. Thus minor differences between cell lines, in response to various combinations of treatments, is highlighted. The data need to be analyzed for statistical significance. - Which doses or time points are taken for comparisons between cell lines? This represents another limitation of quantitation in the paper which reduces ability to interpret results. - Tables presenting p-values don't show the quantitative fold change or percent of cells surviving. These quantitative data need to be added. - Additional cell lines are included at the end of the paper, but they are not studied in early figures. Why was this organized in this manner? All cell lines should be mentioned and studied throughout the study. - Acronyms DNA-PKcs and DNA-PKcsi are not defined. Similarly, other gene names followed by I are not defined. - References to genomics paper about HPV-positive head and neck cancer cell lines and tumors should be included in the citations, such as doi: 10.1101/gr.164806.113 ; DOI: 10.1158/1078-0432.CCR-14-1101; PMID: 30563911. These papers address mutations affecting specific cell lines and disrupting DNA repair pathways. - Choices of inhibitors don't match what is being studied currently in clinical trials. Can the authors explain why they chose not to investigate the particular drugs currently tested in patients ? - images of stained cells show some concerning artifacts. Some wells show stained cell colonies and/ or a haze near the well center, not seen uniformly at the well periphery. Can the authors explain this? - quantitation of the stained cell colonies doesn't appear to match the values shown in the corresponding graphs. How exactly did the authors quantify the cell colonies -- via total density of stain in the well,, by counting discrete colonies regardless of their size, etc. ? The quantitation of the cells is crucial for the paper, and this is unclear.

Reviewer 2 Report

The manuscript is well written. No further comments.

Reviewer 3 Report

Nice study to look into DNA repair in patients afflicted fom head and neck squamous cell carcinoma treated by photons  and prootns, and according HPV-status infection. A very interesting and important topic to look at.

I do not detect relevant limitations or deficiencies in the study.

Reviewer 4 Report

In this manuscript the authors analyze the radiosensitizing effect of the inhibition of ATM, ATR or DNA-PKcs on 2D and 3D cultures of HNSCC cell lines. The conclusions drawn by the authors rely on the analysis of cell survival upon treatment measured via clonogenic assays and spheroid volume evaluation. The results presented are only descriptive, and provide no insight in the molecular mechanisms that underly the differences observed upon the different treatment options. I therefore believe the manuscript is not suitable for publication in Cancers.

Major concerns:

  • The statistic analysis methods used should be described in a dedicated paragraph in the material and method section.
  • Figure 1B-D: it is not clear whether the data presented in this figure represent technical replicates from on single experiment, or mean surviving fractions from several independent experiments. In my experience, clonogenic survival assays can yield variable results from one biologically independent experiment to another. I believe it is crucial that the same tendency is observed over several (N=3 at least) independent experiments in order to draw reliable conclusions. The authors should clarify this issue. In addition, no statistical analysis of the data is provided for the results shown in this figure.
  • Figure 2 and Figure 2: same issues as for figure 1, i.e. how many experiments were perfomed. Please provide a statistical analysis of the data.
  • Table 3 and Table 4: these tables are not easy to understand as they stand: the authors should indicate what variables were compared to generate the ANOVA post-test p-values that are shown.
  • Figure 6: no statistical analysis of the date is provided.
  • One of the most intriguing result of presented by the authors is that inhibition of DNA-Pkcs is relatively uneffective both in 2D and 3D cultures, yet this inhibition seems to me the most effective when combined to photon or protontherapy, compared to the inhibition of ATM or ATR. These observations are not discussed.
  • In general, the data presented is descriptive. The effects of treatments is not analysed using DNA DSBs molecular read-outs for example. It is therefore difficult to compare the radiosensibilisation effects of the treatemnts at a molecular level.

Minor point

  • The authors calculate DERs and show the radiosensitizing effect of drugs combined to photon or proton therapies. Yet, they often use the term "synergy", which as a precise pharmacological definition, and which can usually be calculated with different methods (Chou and Talalay; Bliss score). The use of this term could be misleading to the readers.